# A Dynamic Perspective on the Gender Diversity–Firms' Environmental Performances Nexus: Evidence from the Energy Industry

**Mohamed M. Sraieb \* and Lasha Labadze \***



Finance Department, American University of the Middle East, Egaila 54200, Kuwait
\* Correspondence: mohamed.sraieb@aum.edu.kw (M.M.S.); lasha.labadze@aum.edu.kw (L.L.)

**Abstract:** We explore the role that a country's economic and political uncertainty plays in shaping its environmental performance. We put emphasis on the role played by gender diversity in the board of firms, and we address two limitations characterizing the literature on the topic: (i) the use of static modelling that prevents identifying static and dynamic endogeneity and (ii) the assumption that the relationship is linear, which prevents accounting for the factors that affect the magnitude and the shape of this nexus. Using a System-GMM approach, we find evidence that gender diversity is associated positively with firms' environmental results. Furthermore, the intensity of this relationship is increasing in gender diversity, and more importantly, the effect tends to be greater in less uncertain countries. These findings are of first importance in terms of the policy. Improving environmental quality can be achieved cost-effectively through the promotion of gender diversity, along with building/strengthening institutions to mitigate the effects of economic and political uncertainty. The benefits of these actions can support an effective implementation of the UN SDGs related to gender equality (Goal 5) and several environment-related SDGs (Goal 13 and Goal 15).

**Keywords:** gender diversity; corporate boards; economic and political uncertainty; environmental performance; dynamic endogeneity; system-GMM

## 1. Introduction

The effects of female members in corporate boards have attracted increasing interest, and a stream of literature has investigated the linkages between the share of females in the board of firms and their financial, social, governance, environmental and other performances. In this paper, we concentrate on the firm's environmental results. Two features of previous studies cast doubt on the validity of their findings. First, with very few exceptions, all studies on the topic use static specifications to model the nexus of gender diversity–environmental performances of firms. This makes them unable to address both static and dynamic endogeneity—an inherent problem characterizing this relationship. Very few studies have opted for a dynamic modeling. These include Gaio and Gonçalves (2022) [1], Lu and Herremans (2019) [2], Kassini et al. (2016) [3] and Sila et al. (2016) [4]. Second, most studies implicitly assume a monotonic linear relationship. This is surprising, as it is widely understood that various factors can significantly affect the magnitude and the nature of this relationship. Yet, very few papers have studied these contextual factors. These include the role of a country's development stage (Sraieb and Akin, 2021 [5]), the strength of its institutions, the extent to which the rules of law apply and how well organized its regulatory setting is (Auffhammer and Kellogg, 2011 [6]; Zhang et al., 2018 [7]; Zhao and Luo, 2017 [8]). Noticeably, all these factors have different impacts in different countries. Specifically, the impact varies across countries with different degrees of economic and political uncertainty (Bloom, 2014 [9] and 2009 [10]). As Bloom (2014) [9] puts it clearly, uncertainty is about the expectations of consumers, managers and policymakers regarding the future situation. It also includes doubts concerning the evolution of macroeconomic aggregates, micro-

or firm-level perspectives and global events or shocks (war, climate change, etc.). The literature found evidence that high economic and political uncertainty discourages firms from investing and hiring (Bloom, 2014 [9]), including in abatement activities. Economic policy uncertainty may also push firms to adjust their board composition and consequently affect firms' environmental policy (Ongsakul et al., 2021 [11] have studied this relationship using a static approach). Therefore, a country's economic and political uncertainty status should be explicitly accounted for if the task is to accurately model the connections between the environmental performance of firms and gender diversity in their boards. The main purpose of our paper is to investigate the drivers of the intensity and the shape of the relationship between corporate boards' gender diversity and the environmental performances of firms.

**Hypothesis 1.** *The board's gender diversity positively affects firms' environmental performance.*

In particular, we put weight on countries' uncertainty levels as the contextual factor for this relationship. Therefore, we are testing the following hypothesis:

**Hypothesis 2.** *A country's political and economic uncertainty levels tend to influence the impact of board gender diversity on a firm's environmental performance.*

Our analysis has policy relevance to policymakers, executives and stakeholders. A policy oriented toward promoting the active participation of women in corporate boards would facilitate improved environmental performance. The magnitude of the impact would be even higher in those economies where uncertainty is sufficiently low. The underlying logic is that these tend to be countries with well-established institutions where rule of law is prevailing and the regulatory environment enforced. Therefore, increasing gender diversity in corporate boards of firms in these countries will arguably have a better impact on environmental performance than would occur in countries with a higher level of uncertainty. However, these hypotheses need to be tested using an empirical assessment.

From a methodological perspective, most of the papers use a static model and are thus unable to address dynamic endogeneity. Dynamic endogeneity is an issue when a covariate variable is impacted by past values of the response variable. Intuitively, this is the case of the current framework. This delayed reaction of covariates to the response variables is known as "dynamic endogeneity" (Wintoki et al., 2012 [12]), and if ignored, as is the case in static models, it will result in inconsistent estimates and wrong inferences. This typically occurs when researchers ignore the inherently dynamic nature of a process and fit data with a static specification.

Given these constraints, we choose dynamic modelling—System-GMM—to investigate the correlation between the gender composition in boards of firms and their environmental results. The System-GMM approach is growing in popularity as it leads to consistent estimations (Arellano, 2003 [13]). This approach fits situations with slow-changing independent variables (Antoniou et al., 2008 [14]). Furthermore, it is particularly appropriate for short and wide panels. These are precisely the features of our data.

The contribution of this paper is threefold. First, studies related to this topic focus on one particular country or a region, whereas we analyze the international level (37 countries). Second, unlike most of the literature, we explicitly account for the dynamic feature of the connection between the environmental results of firms and the gender diversity in their boards. Third, our paper accounts for the disparities in the level of economic and political uncertainty across countries and investigates its role in changing the form and the intensity of the relationship under study. We investigate the role of political and economic developments in affecting the environmental performance of firms via promoting gender diversity on boards of firms.

A more gender-diverse board facilitates effective implementation of the UN SDG related to gender equality (Goal 5: "Achieve gender equality and empower all women and girls"), as well as other environment-related SDGs (Goal 13 and Goal 15).

Our paper focuses on the mining, quarrying and oil and gas extraction industry. Along with its magnitude to generate large financial revenues, this industry also exerts strong pressure on the environment and natural resources, as well as on human health (pollution of air, rivers, land, loss of biodiversity, loss of fertile soil, etc.). Scientific reports find that 90% of the loss in biodiversity and constraints on the water is mainly due to the pressure imposed by extractive activities along with the processing phases linked to them. Overall, these activities count for about 50% of the total emissions of greenhouse gases (Oberle et al., 2019 [15]).

Female corporate board members tend to take more ethical business decisions than males related to environmental issues (Lara et al., 2017 [16]). They are also characterized by less risky behavior and tend to be more patient (Lu and Herremans, 2019 [2]) than males. Moreover, females are inclined to care more about the environment than males (Jones and Dunlap 2010). In this paper, we bring international evidence of these findings for the mining, quarrying and oil and gas extraction industry. We also show that the impact is magnified as the country's economic and political uncertainty declines.

The paper proceeds as follows: Section 2 reviews relevant literature on the topic. Section 3 describes the data. Section 4 introduces the settings and discusses the estimation strategy. Our main results are discussed in Section 5. Finally, Section 6 provides concluding remarks.

## 2. Literature Review

Previous studies found that a higher share of female corporate board members improves firms' sets of skills along with their pool of professional experience (Kassini et al., 2016 [3]). Gender diversity in corporate boards also encourages innovative ideas and boosts efficient decision-making processes, as found by Post and Byron (2015) [17]. The same authors conducted a meta-analysis of 87 different studies on the topic and found a significant impact of female board members on the sustainability performance of firms (Byron and Post, 2016 [18]). Furthermore, Lara et al. (2017) [16] found that females are willing to accept more ethical business-related practices, and they are less prone to unethical values. Yarram and Adapa (2021) [19] demonstrated that corporate board gender diversity has a significant impact on firms' CSR. Their findings support both token theory as well as critical mass theory, meaning that only one female director—i.e., token representative—cannot make a significant change, whereas a critical mass of female directors can prevent firms from undertaking negative CSR activities and encourage them to perform more positive CSR activities. This finding is supported by Boukattaya and Omri (2021) [20], who analyze French companies and conclude that board gender diversity is positively associated with CSR and negatively with corporate social irresponsibility (CSI). Females also tend to have a higher aversion to risk compared to men, they are more patient and seek expert views and recommendations before deciding on an uncertain environment, as has been shown by Liu (2018) [21] and Shakil (2021) [22]. Female executives have a higher inclination for empathy and demonstrate more consciousness regarding environmental problems (Jones and Dunlap, 1992 [23]). A very large literature finds significant linkages between the share of females in corporate boards and (i) overall firms' "green" choices (Li et al., 2017 [24]), (ii) the green performances of firms (Lu and Herremans, 2019 [2]), (iii) the environmental sustainability of firms (Glass et al., 2016 [25]), (iv) the disclosure of environmental scores (Ben Ammar et al., 2017 [26]), (v) corporate environmental responsibility (Wang et al., 2021 [27]), etc. Yet, the specific literature on the connection between the share of females in corporate boards and environmental results using contemporary data is rather scarce (Nuber and Velte, 2021 [28]; Sraieb and Akin, 2021 [5]; Cordeiro et al., 2020 [29]; Birindelli et al., 2019 [30]; Zhang et al., 2018 [7]; Zhao and Luo, 2017 [8]). Among these papers, only few make use of a dynamic approach to model this relationship. As discussed above, failure to account for the dynamic nature of that relationship generates inconsistent estimates of the parameters. In order to fix the problem, we follow Gaio and Gonçalves, (2022) [1], Kassini et al. (2016) [3], Sila et al. (2016) [4] and Adams and Ferreira (2009) [31] and we opt

for the use of a dynamic model to take into account the dynamic nature of the relationship. The common feature of all these studies is that they impose a monotonic structure to the relationship between firm performances and gender diversity. This relationship is considered linear, and the intensity of the effect of gender diversity on firms' performances does not depend on the level of gender diversity or any of the other covariates. This is precisely the other limitation we address in our paper.

Few papers have modeled the nexus between gender diversity and firms' performance as a non-linear relationship. These include Sraieb & Akin (2021) [5], who find that the intensity of this relationship depends on the economic development status of the country. Lu and Herremans (2019) [2] find that the strength of this relationship depends on the industry under consideration. More precisely, the effect of gender diversity on firms' environmental performances is stronger in more polluting industries. Birindelli et al. (2019) [30] and Ben-Amar et al. (2017) [26] find evidence for the critical mass hypothesis. The effect of gender diversity in a corporate board materializes only after a threshold number of female directors on the board is reached. We follow these studies and consider a non-linear relationship. To the best of our knowledge, our paper is the only research addressing the moderating role of economic and political uncertainty in a model that accounts for the non-linear dynamic patterns of the environmental performances of firms.

The paper takes the view that higher gender-diverse boards stimulate environmental performance of firms. The magnitude of this relationship is stronger in countries with less economic and policy uncertainty. Furthermore, this magnitude is found to be increasing in gender diversity.

## 3. Data and Descriptive Analysis

The paper traces the environmental performances of 462 firms in the mining, quarrying and oil and gas extraction industry from 37 different countries over 11 years (2008–2018, inclusive). The five largest economies in our data set (Canada, United States, Australia, United Kingdom and China) account for 78% of the firms observed (Figure 1).

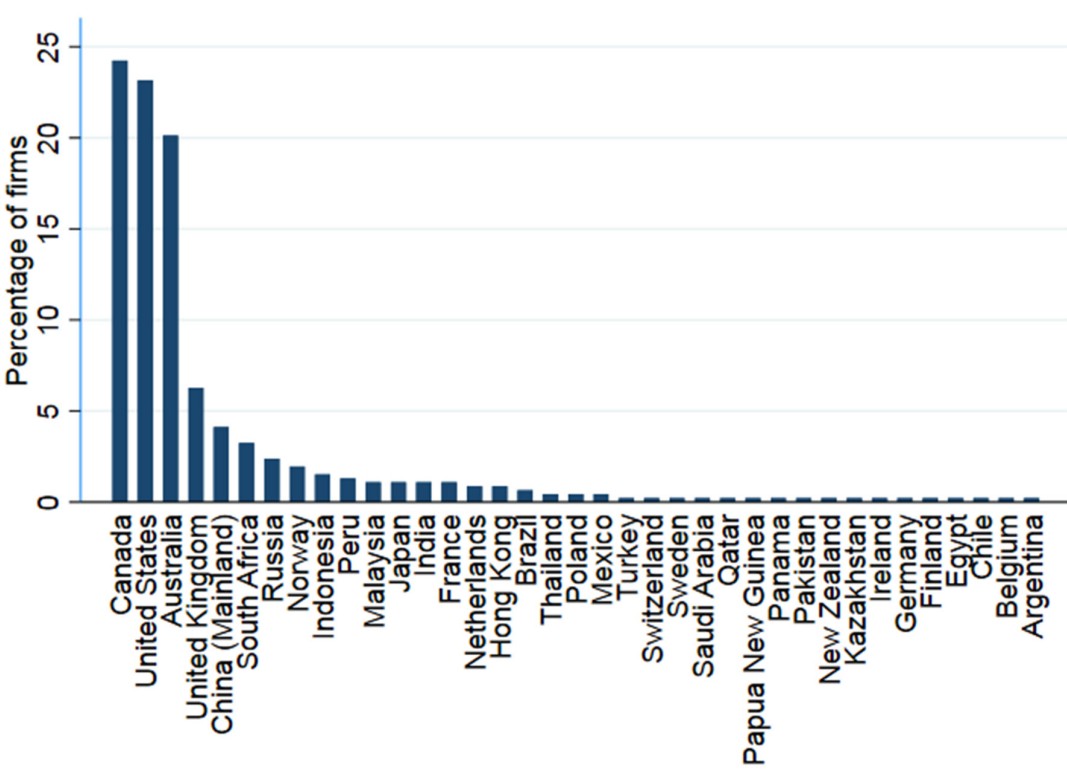

**Figure 1.** Percentage of firms by country.

We follow most of the literature in proxying the environmental performances of companies by the *Environmental Score* provided by Refnitiv Eikon. The environmental Score measures firms' environmental performance. It is the first pillar of the Environmental, Social and Governance (ESG) score from Refinitiv Eikon. It is an overall company score based on self-reported emission, innovation and resources used data by firms. The score ranges from 0 (lowest performance) to 100 (highest performance).

In our model, we use the firm's Environmental Score as a dependent variable, and as a set of independent variables, we include the lagged value of the Environmental Score itself, country's economic and political uncertainty index (WUI), firm level characteristics and indicators of its financial performance: total assets, tangibility, leverage, Tobin's-Q, profitability, gender diversity in corporate board, independence of the board and board's size. We account for individual fixed effects by including unobserved firm-fixed effects and time dummies to control for time-fixed effects, unrelated to the firms' performances. Table 1 provides a description and Table 2 provides a descriptive analysis of all the variables (winsorized at the 1st and 99th percentile) used in our model.

**Table 1.** Description of variables.

| Variable | Description |
|---|---|
| Env.Score | Environmental Score measures firms' environmental performance. |
| WUI | World Uncertainty Index is developed by Hites Ahir, Nicholas Bloom and Davide Furceri (Ahir et al., 2018 [32]). The WUI measures economic and political uncertainty of a country based on Economist Intelligence Unit country reports. The WUI uses a single source for all countries, thereby allowing a comparison of the level of uncertainty across countries. |
| GenDiv | Gender Diversity in corporate boards is measured by the percentage of corporate board seats occupied by females. |
| Firmsize | Firm size is measured as a log of total assets. |
| Tangibility | Net tangible assets (PP&E) to total assets ratio. |
| Leverage | Total debt to total assets ratio. |
| Profitability | Operating profit to total assets ratio. |
| Tobin-q | Tobin's Q is the market value of a firm to total assets ratio. |
| BoardSize | Number of corporate board members. |
| IndpBoard | Share of independent board members. |

**Table 2.** Summary statistics. All variables are winsorized at the 1st and 99th percentile to minimize outliers' effects.

| | Mean | Median | St.Dev | Min | Max | N |
|---|---|---|---|---|---|---|
| Env.Score | 46.751 | 45.07 | 19.889 | 9.01 | 95.227 | 3610 |
| WUI | 0.07 | 0.06 | 0.051 | 0 | 0.418 | 5082 |
| GenDiv | 0.096 | 0.083 | 0.110 | 0 | 0.5 | 3605 |
| Firmsize | 21.121 | 21.292 | 2.009 | 15.403 | 25.605 | 4767 |
| Tangibility | 0.591 | 0.63 | 0.244 | 0 | 0.946 | 4704 |
| Leverage | 0.21 | 0.187 | 0.195 | 0 | 0.979 | 4753 |
| Profitability | 0.036 | 0.045 | 0.149 | −1.126 | 0.419 | 4717 |
| Tobinq | 1.601 | 1.255 | 1.172 | 0.457 | 11.094 | 4433 |
| BoardSize | 8.628 | 8 | 2.981 | 4 | 21 | 3605 |
| IndpBoard | 0.631 | 0.667 | 0.227 | 0 | 1 | 3585 |

The main focus of the analysis will be on the two explanatory variables WUI and GenDiv. The former represents our measure of uncertainty, the latter is the percentage of females in the boards of firms. The percentage of corporate board seats occupied by females was increasing steadily from 6% in 2008 to 15% in 2018 (Figure 2a).

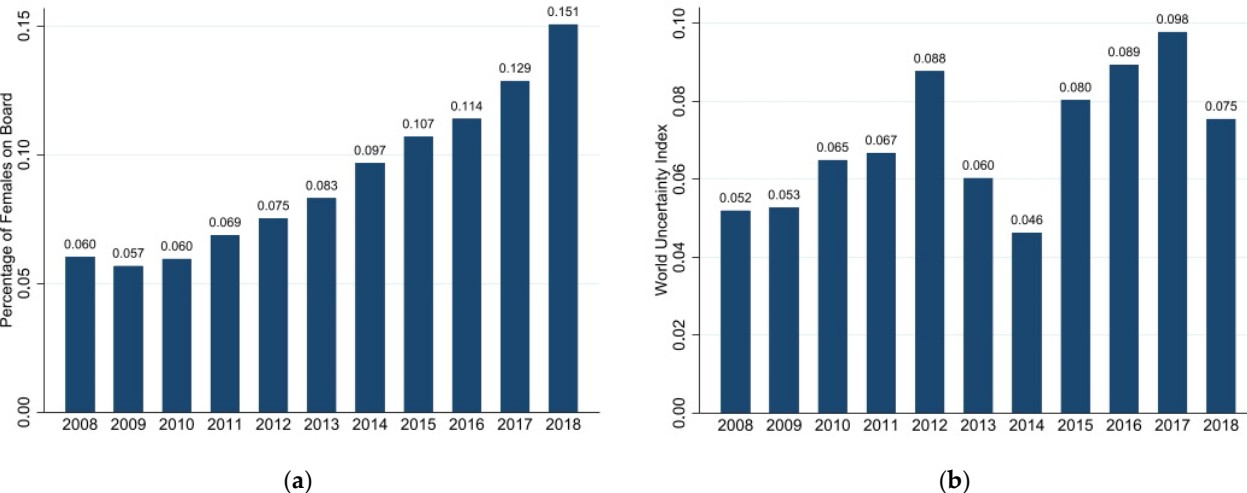

(**a**)                                                                 (**b**)

**Figure 2.** (**a**) Share of females in corporate boards for the mining, quarrying and oil and gas extraction industry and (**b**) World Uncertainty Index, 2008–2018.

The average of corporate boards' gender diversity for mining, quarrying and oil and gas extraction industry during 2008–2018 was about 9.6%. The pattern looks promising, but it is still far from a gender balance in corporate boards of firms. Unlike gender diversity in corporate boards, the WUI demonstrates a cyclical pattern ranging from 5% to 10% (Figure 2b). The higher value of WUI indicates higher economic and political uncertainty in a country, which might be caused by weak institutions, a weak regulatory framework and rule of law. For analysis purposes, we first generate a dummy variable (WUI dummy), which is equal to 1 if a country's WUI is above the median for a given year and 0 otherwise.

As discussed in the introduction, the focus on the mining, quarrying and oil and gas extraction industry is dictated by the intensity of its negative impact on the environment. This industry has historically been male-dominated, therefore, the impact of an improved female share in boards of firms on their environmental results in this particular industry would be of utmost importance.

## 4. Model Specifications

The paper uses a dynamic panel data model to assess how gender diversity affects firms' environmental decisions:

$$\text{Env.Score}_{it} = \alpha_i + \delta_t + \beta\,\text{Env.Score}_{it-1} + X'_{it-1}\vartheta + \varepsilon_{it};$$

$$i = 1, \ldots, 462; \; t = 1, \ldots, 10$$

(1)

with Env.Score represents environmental score of firm i in year t. Individual and time fixed effects are represented by $\alpha_i$ and $\delta_t$, respectively. We conjecture that the relationship under investigation is dynamic, by essence. Therefore, we add the lagged dependent variable Env.Score$_{it-1}$ as an additional covariate. The matrix $X_{it-1}$ groups all other covariates lagged one year to allow for the delay in transmitting the impact. This matrix includes also financial ratios as control variables. These include leverage, Tobin's Q, profitability, firm size, etc. Finally, $\varepsilon_{it}$ is the error term, with $\varepsilon_{it} \sim \text{iid}(0, \sigma)$.

From an econometric perspective, the rationale for lagging all explanatory variables is to minimize the extent of simultaneity. Moreover, to address skewness in some variables,

we apply the natural log whenever possible (absence of null values). All variables are winsorized at the 1st and 99th percentile to minimize outliers' effects.

Having the lagged environmental score (Env.Score$_{it-1}$) as a covariate in (1) would make the fixed effect estimator biased and inconsistent. Indeed, E($\alpha_i$ | Env.Score$_{it-1}$) $\neq$ 0; therefore, within transformation does not solve the problem, as it generates a correlation between Env.Score$_{it-1}$ and the differentiated $\varepsilon_{it}$. Moreover, simultaneity may be an issue in our particular case as one cannot dismiss the idea that on the one hand, more females in corporate boards may lead to better environmental records for a firm. On the other hand, better environmental records may attract more females to firms' boards. Endogeneity is a typical issue in virtually all studies in the field (Adams and Ferreira, 2009 [31]; Campbell and Minguez-Vera, 2008 [33]). This puts a limit on the validity of inferences and causality (Wintoki et al., 2012 [12]). The problem is all the more difficult than finding a "suitable" instrument for gender diversity is a challenge in itself (Adams and Ferreira, 2009 [31]).

Therefore, we opt for the use of the System-GMM approach implemented by Arellano and Bover (1995) [34] and conceived by Blundell and Bond (1998) [35]. The method generates a higher finite sample properties estimator (Blundell et al., 2001 [36]). Among the salient features of the System-GMM approach is that it effectively addresses both of the problems discussed above. It addresses the endogeneity concerns by using internal instruments (Holtz-Eakin et al., 1988 [37]; Blundell and Bond, 1998 [35]; Arellano and Bond, 1991 [38]; Arellano and Bover, 1995 [34]), and by the same token, it accounts for the dynamic nature of (1). Adopting the lags of the endogenous variables in levels and differences as internal instruments seems too appealing and simplistic. However, its implementation may lead to a serious problem, namely, "instrument proliferation" (Roodman, 2009 [39]). The problem consists of over-fitting endogenous variables and therefore generating high false-positive results for the Hansen J-test, validating instruments that otherwise would not be retained as valid.

To avoid bias and inconsistency that might be caused by instruments proliferation, we collapse the instruments matrix. To reiterate, this econometric technique is exceptionally appropriate for short and wide panels, which exactly fits our case.

## 5. Findings and Discussion

The estimation of the baseline model, where $\beta$ = 0 in Equation (1), which makes it a fixed effect, is reported in the first column of Table 3. A one percentage point (0.01) rise in the lagged GenDiv increases the firm's environmental score (Env.Score) by about 0.21%, ceteris-paribus. The result is significant at the 5% level. Two other statistically significant variables are the lag of Firmsize (firm size) and IndpBoard (board independence). They are both positively correlated to the environmental performances of a firm. The results are in-line with the findings of previous studies (Antara et al., 2020 [40]; Younis and Sundarakani, 2019 [41]). The baseline approach is criticized on the ground endogeneity concerns (Adams and Ferreira, 2009 [31]; Lu and Herremans, 2019 [2]). Intuitively, a higher share of females in corporate boards may affect a firm's environmental performance, but at the same time, firms with better environmental performances may be more attractive to female managers. Self-selection bias may be an issue leading to biased and inconsistent estimates.

We address this issue using a dynamic panel data model and introducing a lagged dependent variable (column 2, Table 3). As discussed above, we implement a system-GMM model. This particular panel data method is chosen over its ability to address the endogeneity issues generated by the introduction of gender diversity as an independent variable (Harris and Matyas, 2004 [42]), and by the same token, addresses the dynamic panel bias.

Following column 2 of Table 3, a 1 percentage point increase in gender diversity in corporate board improves a firm's environmental performance in the following year by about 0.2%, holding other factors constant. The result is statistically significant at the 5% level, and it is consistent with other specifications of the System-GMM model, as shown in columns 3 and 4 of Table 3. The magnitude of this coefficient is better demonstrated

if we consider a firm with eight corporate board members (the median value of board size, Table 2) with two female members among them initially. Consequently, the share of females is equal to 0.25. If the share of females increases from 0.25 to 0.50, or 25 percentage points (from two to four female board members), then the environmental performance score would increase by 25 × 0.2% = 5% on average, ceteris paribus.

**Table 3.** Effects on Environmental Score, FE and GMM models.

|  | (1) | (2) | (3) | (4) |
|---|---|---|---|---|
| L.GenDiv | 0.213 ** | 0.184 ** | 0.181 ** | 0.187 ** |
|  | (0.100) | (0.0791) | (0.0789) | (0.0793) |
| L.firmsize | 0.112 *** | 0.0150 ** | 0.0158 ** | 0.0162 ** |
|  | (0.0175) | (0.00627) | (0.00634) | (0.00652) |
| L.Leverage | −0.0180 | −0.0373 * | −0.0370 * | −0.0354 * |
|  | (0.0619) | (0.0217) | (0.0216) | (0.0215) |
| L.logtobinq | 0.0242 | 0.0165 ** | 0.0167 ** | 0.0157 ** |
|  | (0.0212) | (0.00669) | (0.00670) | (0.00662) |
| L.profitability | 0.0766 | 0.0462 | 0.0502 * | 0.0516 * |
|  | (0.0794) | (0.0301) | (0.0303) | (0.0297) |
| L.tangibility | −0.0110 | −0.0120 | −0.0132 | −0.0156 |
|  | (0.0527) | (0.0189) | (0.0189) | (0.0183) |
| L.logBoarSize | 0.0657 | −0.0155 | −0.0131 | −0.0106 |
|  | (0.0427) | (0.0156) | (0.0159) | (0.0159) |
| L.IndpBoard | 0.0979 ** | −0.00504 | −0.00119 | −0.00398 |
|  | (0.0412) | (0.0162) | (0.0165) | (0.0176) |
| L.logEnv.Score |  | 0.897 *** | 0.893 *** | 0.888 *** |
|  |  | (0.0475) | (0.0477) | (0.0483) |
| WUI_Dummy |  |  | −0.0122 * |  |
|  |  |  | (0.00727) |  |
| WUI_Q = 2 |  |  |  | −0.0117 |
|  |  |  |  | (0.0109) |
| WUI_Q = 3 |  |  |  | −0.00178 |
|  |  |  |  | (0.00931) |
| WUI_Q = 4 |  |  |  | −0.0187 * |
|  |  |  |  | (0.0105) |
| Constant | 1.202 *** | 0.121 * | 0.125 * | 0.132 ** |
|  | (0.369) | (0.0657) | (0.0662) | (0.0632) |
| N | 2964 | 2964 | 2964 | 2964 |
| N of Instruments |  | 27 | 28 | 30 |
| AR1 $p$ value |  | 0.000 | 0.000 | 0.000 |
| AR2 $p$ value |  | 0.380 | 0.401 | 0.408 |
| Hansen J-test $p$-value |  | 0.184 | 0.211 | 0.218 |
| Hansen diff $p$-value |  | 0.108 | 0.129 | 0.146 |

Standard errors in parentheses. Dependent Variable is log of Env.Score. * $p < 0.10$, ** $p < 0.05$, *** $p < 0.01$.

In the following specification of the System-GMM model (Table 3, column 3), we add a dummy variable of WUI defined as 1 if the country's WUI is above its median value and as 0 otherwise in a given year. The coefficient of the WUI dummy is negative and significant at the 10% level, indicating that more uncertain countries have lower environmental performance scores by about 1.2%. We further investigate the effect of WUI on a firm's green performance by including dummies for quartiles. For example, WUI Q = 2 is the dummy variable for the second quartile, meaning that the corresponding WUI falls between the 25th and 50th percentiles. We observe that the only statistically significant difference is detected in the highest quartile, where WUI is above the 75th percentile. This finding suggests that those countries that fall in that range of uncertainty have a statistically significant (at the 10% level) poorer environmental performance by about 1.9%.

The coefficient on the lagged Env.Score variable for all three System-GMM regressions in Table 3 (columns 2–4) indicates that environmental score has a large coefficient pointing

to a high persistence, meaning that a firm's environmental performance during a certain year highly influences its environmental score in the following year (see coefficient of L.logEnv.Score, Table 3). Typically, a 1% change in the environmental performance of a firm in a particular year corresponds to a 0.9% change in the next year performance, holding other factors constant. This finding comforts our idea that the connection between the share of women in boards of firms' and their records in terms of green performance is dynamic, in essence. This calls naturally for the implementation of a dynamic framework.

Results in Table 3 also indicate that *firm size* has positive association with the environmental performance score across all models. The result is significant at the 5% level, at least. This finding is in line with those from a previous study (Lu and Herremans, 2019 [2]). Firms' financial performance, measured by Tobin's Q, has a positive coefficient and the result is significant at the 5% level. Having a higher Tobin's Q predicts better environmental scores for firms in the following year.

The environmental performance of any firm in a given country would depend on the existence of well-functioning strong institutions and on the strictness of regulations, its economic and political stability, as well as on its people's consciousness of the environmental issues. The results in Table 3 also reveal that gender diversity in firms' board and countries' uncertainty scores are important determinants of firms' environmental performance. However, the magnitude of the impact of gender diversity differs across countries with different economic and political uncertainty levels. Therefore, we hypothesize that a country's political and economic uncertainty level, as measured by WUI, tends to influence the impact of board gender diversity on a firm's environmental performance. To further investigate this phenomenon, we introduce the interaction of the WUI dummy and the GenDiv variable in the model, and we try two specifications first: (a) allow only the slope coefficient to change (Table 4, column 1) and (b) allow the intercept and slope coefficients of these explanatory variables to change (Table 4, column 2). In both specifications, the coefficients GenDiv and its interaction term with the WUI dummy are significant at the 5% level. The sign of the coefficient of the interaction indicates that the impact of gender diversity is lower in countries with higher uncertainty.

Greater gender diversity for a firm located in a country with high political and economic uncertainty would lower the firm's environmental performance ($0.616 - 0.853 = -0.237$). More precisely, a 1 percentage point increase in GenDiv in a country with high political and economic uncertainty leads to an average decline of 0.24% in the firm's environmental performance in the following year.

This suggests that the previous finding on the positive relationship between more gender-diverse boards and a firms' environmental performance tends to be conditional on reduced policy and economic uncertainty. More gender-diverse corporate boards would not lead to better environmental performances of firms in countries with higher uncertainty (higher WUI score). The rationale for this result stems from the idea that uncertainty adversely affects the enabling environment of an economy. This may induce financial outcomes for firms that are so negative and diffuse that they could hardly be balanced by the benefits brought with more gender-diverse boards (Bloom, 2014 [9]; Atsu and Adams, 2021 [43]). Indeed, uncertainty hampers household confidence and makes it difficult for businesses to plan for the future. The lack of visibility for future economic and political prospects undermines agents' confidence and further raises their aversion to risk.

In terms of policy, our finding suggests that promoting gender diversity on corporate boards should be accompanied by measures that reduce political and economic uncertainty. This is particularly relevant for countries vulnerable to shocks whether internal or external (COVID-19, war, conflicts, trade tensions, etc.). These can further exacerbate agents' risk-aversion and deteriorate the market ability to create a sound economic and political environment in which agents interact effectively.

**Table 4.** Effects on Environmental Score, FE and GMM models with WUI.

| | (1) | (2) | (3) | (4) |
|---|---|---|---|---|
| L.logEnv.Score | 0.874 *** (0.0463) | 0.912 *** (0.0512) | 0.878 *** (0.0465) | 0.887 *** (0.0500) |
| L.firmsize | 0.0190 *** (0.00588) | 0.0127 * (0.00691) | 0.0186 *** (0.00615) | 0.0164 ** (0.00667) |
| L.Leverage | −0.0405 * (0.0223) | −0.0471 ** (0.0236) | −0.0359 * (0.0217) | −0.0352 (0.0230) |
| L.logtobinq | 0.0162 ** (0.00708) | 0.0182 *** (0.00700) | 0.0168 ** (0.00682) | 0.0190 *** (0.00694) |
| L.profitability | 0.0555 * (0.0310) | 0.0253 (0.0337) | 0.0569 * (0.0302) | 0.0497 (0.0307) |
| L.tangibility | −0.0221 (0.0183) | −0.0153 (0.0192) | −0.0195 (0.0179) | −0.0159 (0.0184) |
| L.logBoarSize | −0.00584 (0.0163) | −0.00406 (0.0150) | −0.0105 (0.0154) | −0.0106 (0.0155) |
| L.IndpBoard | 0.00285 (0.0168) | 0.00761 (0.0171) | −0.00218 (0.0171) | −0.00117 (0.0180) |
| WUI_Dummy = 1 × L.GenDiv | −0.162 ** (0.0796) | −0.853 ** (0.382) | | |
| L.GenDiv | 0.184 ** (0.0784) | 0.616 ** (0.247) | 0.206 *** (0.0767) | 0.380 ** (0.182) |
| WUI_Dummy = 1 | | 0.0674 * (0.0363) | | |
| WUI_Q = 2 × L.GenDiv | | | −0.163 (0.116) | −0.327 (0.336) |
| WUI_Q = 3 × L.GenDiv | | | −0.0293 (0.107) | −0.0727 (0.446) |
| WUI_Q = 4 × L.GenDiv | | | −0.165 * (0.0872) | −0.548 * (0.282) |
| WUI_Q = 2 | | | | 0.0151 (0.0327) |
| WUI_Q = 3 | | | | 0.00252 (0.0395) |
| WUI_Q = 4 | | | | 0.0463 (0.0316) |
| Constant | 0.124 * (0.0679) | 0.0645 (0.0749) | 0.121 * (0.0632) | 0.112 * (0.0670) |
| N | 2964 | 2964 | 2964 | 2964 |
| N of Instruments | 29 | 29 | 33 | 33 |
| AR1 *p* value | 0 | 0 | 0 | 0 |
| AR2 *p* value | 0.440 | 0.519 | 0.445 | 0.438 |
| Hansen J-test *p*-value | 0.215 | 0.518 | 0.340 | 0.341 |
| Hansen diff *p*-value | 0.233 | 0.309 | 0.267 | 0.228 |

Standard errors in parentheses. The dependent variable is log of Env.Score. * $p < 0.10$, ** $p < 0.05$, *** $p < 0.01$.

Number of instruments (27–33) across models, as reported in Tables 3 and 4, indicate that the proliferation of instruments (Roodman, 2009 [39]) is not a concern in our estimations. The serial correlation AR(1) test p-values are close to zero and the AR(2) test p-values are in the 0.380–0.519 range across different specifications in both tables, suggesting that the null

hypothesis of the absence of serial correlation of the second order for disturbances in the first difference equation cannot be rejected. Furthermore, the Hansen J-test (0.184–0.518) does not reject the validity of instruments as a group. Furthermore, Tables 3 and 4 reports on the Hansen J-test in differences. This refers to the exogeneity of the subsets of instruments. The corresponding *p*-values (0.108–0.309) do not reject the null hypothesis of exogeneity of these subsets, bringing further consolidation for robustness of our results.

In order to assess which level of the country's uncertainty induces the negative effect of gender diversity, we introduce interaction terms with WUI quartile dummies (again, with two specifications, allowing slope and then the both slope and intercept to change). The findings are summarized in Table 4 (columns 3 and 4). They suggest that the interaction term with only the last quartile is statistically significant (at the 10% level), indicating that the impact of GenDiv on environmental performance is mostly driven by highly uncertain countries (those in the last quartile of the distribution). Countries below the 75th percentile of the WUI do not exhibit statistically significant differences among each other.

We investigate this finding further by examining, for different gender diversity levels, (a) the intensity of the relationship across high and low uncertainty levels and (b) the marginal impact of a country's uncertainty status on the firm's environmental performance (Figure 3). We find that in countries with a high (above median) uncertainty level, the environmental performances of firms are significantly lower than in countries with low uncertainty, and this impact is magnified as gender diversity in corporate boards increases. The finding is robust for both specifications, with fixed and without fixed intercepts. Higher gender diversity in countries with lower uncertainty increases a firm's environmental performance, whereas, in countries with high uncertainty, an increase in female board members has a negligible positive impact (Figure 3a) or even a negative impact (Figure 3c). The gap in environmental performance between countries with high versus low uncertainty increases as gender diversity in corporate boards increases.

A potential explanation of this finding refers to the exacerbating effects of political and economic uncertainties on entrepreneurs' risk-aversion. This, in turn, discourages investments in costly abatements technologies and environmentally friendly processes. More generally, increased risk aversion reduces investments and activities whose returns accrue more in the long-run. These include research and development effort, and particularly environmental-led activities, which are seen as secondary in firms' scale of priorities. Ultimately, this would put pressure on firms' behavior and worsen their environmental performances (Bloom, 2014 [9]; Atsu and Adams, 2021 [43]).

The arguments above provide a rationale for the worsened environmental performances of firms in response to higher uncertainties and explain the gap in effects for firms across countries of different uncertainty levels. However, this does not address the potential reasons behind the widening of this gap for higher gender diversity levels. One potential explanation of this result relates to the particular attitude of women toward risk. Females on boards tend to exhibit more risk aversion compared to their male peers. They tend to show more patience and are typically more willing to look for professional guidance when facing uncertainty (Liu, 2018 [21]). Females on boards tend to be more inclined to wait until the uncertainty is resolved before making major business decisions. Therefore, the more females on corporate boards, the greater the weight of their decisions (i.e., the higher would be the effect of uncertainty in firms' decisions).

This finding begs the question of whether the differential between countries with high versus low uncertainly levels is statistically significant for increasing gender diversity levels. Figure 3b,d) trace the magnitude of the gap between the low and high uncertainty scores. It states that this gap is increasing, in absolute value, and is statistically significant at the 95% level for all values of gender diversity (for gender diversity levels larger than 10%, for both specifications Figure 3b,d). This threshold value points to a standard and very important result in the literature. This refers to the critical mass theory, by which the impact of gender diversity on the environmental performance of firms materializes only when a critical mass weight of females is realized in corporate boards (Konrad et al.,

2008 [44]; Torchia et al., 2011 [45]; Schwartz-Ziv, 2017 [46]; among others). This suggests that an effective way to improve environmental quality and mitigate the adverse effects of economic activities is to encourage gender diversity in corporate boards in countries with low uncertainty where we find a higher impact on the firms' environmental performances.

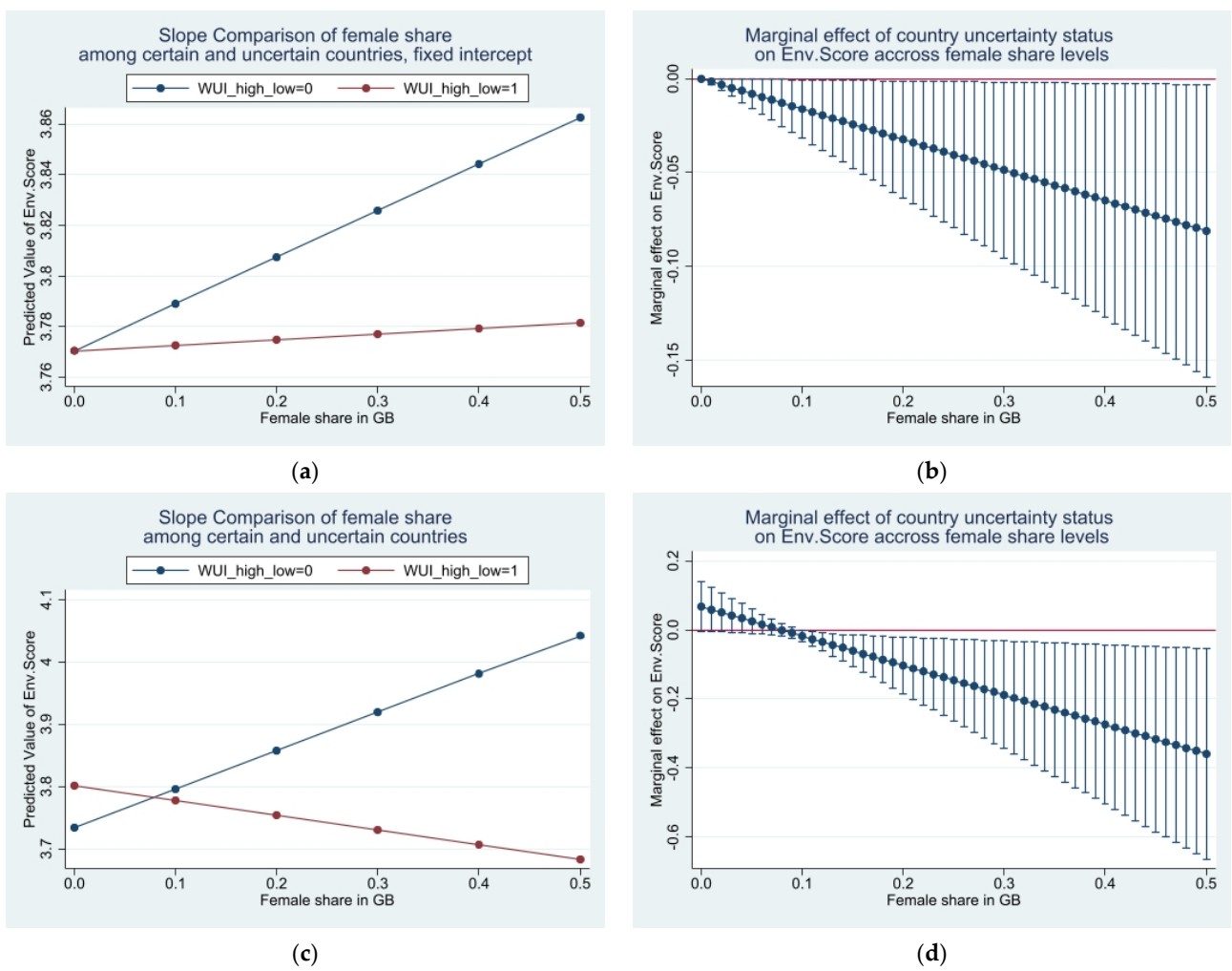

**Figure 3.** (**a**,**c**) Slope comparison of the female share among countries with high and low uncertainty; (**b**,**d**) corresponding marginal effects of the country's uncertainty level on environmental performance across different levels of the female share in corporate boards.

In terms of policy, our findings suggest that promoting gender diversity cannot reach full potential, and may even lead to adverse results unless accompanied by measures that mitigate political and economic uncertainty. These measures would create an enabling environment both for households and entrepreneurs. These reduce their aversion to risk, which increases their confidence and willingness to invest. To unlock this potential, a country should implement accompanying measures that promote and strengthen stable institutions–as a defense against uncertainty. They contribute to anchoring economic agents' expectations and stabilize the economy.

## 6. Conclusions

Environmental issues are one of the most challenging problems today, as they manifest in many different ways and affect the well-being of humans. This is witnessed by the overwhelmingly high number of UN-SDG indicators devoted to the environment. Strikingly, 93 of the 244 indicators of the UN-SDG framework are environment-related. Thus, an understanding of the determinants of the environmental performance of firms is

of utmost importance. Using dynamic modelling, this paper sheds light on the positive effect of gender diversity in corporate boards on a firm's environmental performance in different countries.

Our results confirm that the magnitude of the effect is determined by the level of political and economic uncertainty characterizing a country. The impact of a more gender-diverse board on the green performance of a typical firm is stronger in less uncertain countries. Overall, firms with a higher gender diverse board record better environmental results. Nevertheless, in highly uncertain countries (with WUI above the 75th percentile), the impact of this relationship is found to be negligible or even negative.

Furthermore, we find that impact of gender diversity on environmental performance is statistically different between certain and uncertain countries if female board members are above 10%, which is in line with critical mass theory, by which the effects of improving gender diversity materialize only after the number of females on corporate boards is sufficient to ensure their weight into the board decisions is high enough to be impactful.

From a policy perspective, the emphasis put on gender diversity by international organizations, scholars, researchers and practitioners must be seen as leverage for improving firms' performances (environmental, financial, governance, social responsibility, etc.). Our findings suggest that the effectiveness of such a gender-led policy is not homogenous across countries with different political and economic uncertainty levels. Therefore, improving the effectiveness of gender diversity policies goes through mitigating uncertainties. This can best be achieved through building and empowering institutions. These would create an enabling environment both for households and entrepreneurs. Institutions mitigate risk aversion, increase agents' confidence, dump their willingness to invest and expand activity.

The finding of the critical mass theory also suggests that a very effective way to improve environmental quality and mitigate the adverse effects of economic activities is to encourage gender diversity in corporate boards in countries with low uncertainty where we find a higher impact on firms' environmental performances. In these countries, the margin for progress is large and the effect of improving gender diversity in corporate boards materializes more easily and at lower costs, as these countries are far from their efficiency frontiers.

**Author Contributions:** Conceptualization, M.M.S.; Data curation, L.L.; Formal analysis, L.L.; Methodology, M.M.S.; Project administration, M.M.S. and L.L.; Resources, M.M.S.; Software, M.M.S. and L.L.; Supervision, M.M.S. All authors have read and agreed to the published version of the manuscript.

**Funding:** This research received no external funding.

**Data Availability Statement:** Restrictions apply to the availability of these data. Data was obtained from Refinitiv Eikon and are available at https://eikon.refinitiv.com with the permission of Refinitiv Eikon.

**Conflicts of Interest:** The authors declare no conflict of interest.

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
