# Peer review of "A Dynamic Perspective on the Gender Diversity–Firms’ Environmental Performances Nexus: Evidence from the Energy Industry"

_sustainability, doi:10.3390/su14127346_

Round 1

Reviewer 1 Report

I believe that, despite being a topic that is widely discussed in the literature, it is interesting to be able to provide new orientations, in this case the gender diversity firms’ environmental performances . I see the work interesting, although it needs a good reorientation, in order to make it more attractive to reading. I consider the paper can bring interesting contributions in various fields. However, I will make some suggestions, so that it can be improved. 

1- need to update the literature review with recent studies

2- Please add a theoretical background section

3- What are the policy implications of your study

4- Add discussion of results section and relate your results with theories. 

Author Response

Dear Reviewer,

Many thanks for your comments and suggestions. Your review truly helped improve further the paper. Below we provide the details on how your comments and suggestions have been addressed. We take this review as a valuable opportunity to foster the quality of the paper.

1- Need to update the literature review with recent studies

We added more coherent theoretical arguments to the discussion based on the most recently published relevant academic papers on the topic (lines 125-133, 153-158, 442-452). The incorporated studies include Yarram and Adapa (2021), Boukattaya and Omri (2021), Shakil (2021), Wang, Wilson & Li, (2021), Cordeiro et al 2020, Birindelli, et al. (2019), and Nuber & Velte (2021). In total, the list of references increased from 21 to 36. The details of these studies and the logic behind their inclusion are explained in the text (pages 3 and 4).

For the quantitative papers, our guiding line is the inclusion of papers proposing a reliable strategy to address the problem of endogeneity characterizing virtually all studies dealing with gender diversity. In a nutshell, gender diversity and firms’ performances (be they financial, ESG or environmental) have reverse causality. More gender-diverse boards are found to improve firms’ performances, and at the same time, better performing firms are more attractive to females. Any study incorporating these two variables would suffer from endogeneity. Therefore, inference and results are biased. Finding an instrument for gender diversity becomes a concern of first importance. The task is challenging as all potential candidates for being a suitable instrument should be already part of the regression equation. So far, the literature has not found an instrument unanimously agreed upon. This is the reason for which our selection of papers is rather limited. We focused on those papers with a choice of instrumental variables that are well explained and papers that ‘’go around’’ the problem by making use of a method that does not rely on external instruments. This includes papers using difference-GMM or system-GMM estimators. Their principal advantage is the use of internal instruments instead of external ones. Indeed, the instruments chosen are difference and the level of the covariates themselves. There are only a few of these studies available, including ours. This again explains the relatively restricted number of references we are considering.

2- Please add a theoretical background section

Initially, we opted for integrating the theoretical background disseminated throughout the text to connect to the findings of related papers and to avoid redundancy and repetition. Changes that we incorporated in response to this comment improve the overall theoretical framework of the paper, including concepts, definitions and links to relevant literature. We also articulated theoretical assumptions and added a comprehensive discussion of the observed phenomena. Specifically,  we added critical mass theory discussion in the literature review section (lines 125-132). We added interpretation of corresponding results in the findings and discussions section (lines 445-452) and the conclusion section (lines 482-503). These changes in addition to the existing discussions of theories and concepts significantly improve the theoretical background of the paper.

3- What are the policy implications of your study

We answered this comment along different dimensions. We revise and elaborate on the explanation and interpretation of the results. We provide further explanations on the ways political and economic uncertainty shapes the relationship between corporate board gender diversity and firms’ environmental performance. We also emphasize the policy implications of these findings (lines 453-462, 497-503). To better reflect the policy implications of the study, we incorporate changes that are diffuse through the text. In particular, we argue that promoting gender diversity cannot deliver its full potential, and may even lead to adverse results unless it is accompanied by measures that mitigate political and economic uncertainty. These measures would create an enabling environment both for households and entrepreneurs. Such measures should increase agents’ confidence, dump their willingness to invest and expand the activity by reducing their aversion to risk. The accompanying measures should primarily target building and strengthening institutions to unlock this potential. Institutions are a defence against uncertainty. They contribute to anchoring economic agents’ expectations and stabilize the economy.

We also elaborate on the finding of the critical mass theory. This result suggests that a very effective way to improve environmental quality and mitigate the adverse effects of economic activities is to encourage gender diversity in corporate boards in countries with low uncertainty where we find a higher impact on firms’ environmental performances. In these countries, the margin for progress is large and the effect of improving gender diversity in corporate boards materializes more easily and at lower costs, as these countries are far from their efficiency frontiers (lines 445-461 and 482-503).

4- Add a discussion of results section and relate your results with theories. 

We addressed your suggestion to improve arguments and discussion of findings by revising existing and adding new paragraphs better explaining regression coefficients that demonstrate how political and economic uncertainty shapes the relationship between corporate board gender diversity and firms’ environmental performance. (Lines 379-389).

Furthermore, we added supporting arguments disseminated across the text (particularly in lines 348-361, lines 379-389, etc). These relate our finding to theory. We focus here on the attitude towards risk and uncertainties. The related theory posits that females’ attitude when facing uncertainty is different from males. Women on corporate boards tend to be more risk-averse and lean more toward conservatism in uncertain environments. They tend to exhibit a strong inclination to wait until the uncertainty is resolved before making major business decisions. Therefore, the more females there are on corporate boards, the higher their weight in the board decisions would be, and by the same token, the higher the effect of uncertainty on firms’ decisions.

Moreover, we also relate our findings to the critical mass theory, suggesting that the impact of gender diversity on the environmental performances of firms materializes only when a critical mass weight of females is realized in corporate boards (Konrad et al. 2008, Torchia et al. 2011, Schwartz-Ziv, 2017- page 12, lines 445-451).

Reviewer 2 Report

Thank you for your hard work.

This is a well-executed study with a solid research base.

I only have a few minor concerns that I'd like the authors to clarify and address when revising.

  • "To the best of our knowledge, this is the only study that addresses the role of economic and political uncertainty as a moderator in a model that accounts for the dynamic patterns of firm environmental performance" (line 128). --> I believe the authors intended "moderating role."
  • The authors could go into greater detail to explain the logic behind this relationship. As it stands, the paper offers little insight into the reasoning behind their hypothesis.
  • It would be useful if the authors provided some speculative thoughts on the finding that the slope of the GenDiv variable is negative in highly uncertain countries on the firm's environmental performance (Figure 3 (c) red line).

Overall, I believe that this study has a high potential to add to the literature on the relationship between gender diversity and firm environmental performance. Best wishes for your future research.

Author Response

Dear Reviewer,

Many thanks for your comments and suggestions. Your review truly helped improve further the paper. Below we provide the details on how your comments and suggestions have been addressed. We take this review as a valuable opportunity to foster the quality of the paper.

The only remark on the content refers to “Are the arguments and discussion of finding coherent, balanced and compelling? Can be improved”.  The reviewer kindly gives details of the elements that can be improved in the field ‘’Comments and Suggestion for Authors’’. We address each of them as follows:

  1. "To the best of our knowledge, this is the only study that addresses the role of economic and political uncertainty as a moderator in a model that accounts for the dynamic patterns of firm environmental performance" (line 128). --> I believe the authors intended "moderating role."

Absolutely. Changed accordingly (line 160).

  1. “The authors could go into greater detail to explain the logic behind this relationship. As it stands, the paper offers little insight into the reasoning behind their hypothesis.”

We answered this comment along different dimensions. We revise and elaborate on the explanation and interpretation of the results. We provide further explanations on the ways political and economic uncertainty shapes the relationship between corporate board gender diversity and firms’ environmental performance. We also emphasize the policy implications of these findings (lines 453-461, 487-496).

In addition, we add more theoretical arguments to the discussion based on the most recently published academic papers on the topic. (Lines 421-452)

  1. “It would be useful if the authors provided some speculative thoughts on the finding that the slope of the GenDiv variable is negative in highly uncertain countries on the firm's environmental performance (Figure 3 (c) red line).”

To address this comment, we added supporting arguments disseminated across the text (particularly in lines 348-361, lines 421-428, etc). These explain the finding you are referring to by attitude towards risk and uncertainties. The related theory posits that females’ attitude when facing uncertainty is different from males. Women on corporate boards tend to be more risk-averse and lean more toward conservatism in an uncertain environment. They tend to exhibit a strong inclination to wait until the uncertainty is resolved before making major business decisions. Therefore, the more females there are on corporate boards, the higher would be their weight in the board decisions, and by the same token, the higher would be the effect of uncertainty on firms’ decisions.

Reviewer 3 Report

Dear Authors

The paper has been significantly improved since the previous version.

Yet, some issues still need to be addressed:

1- In lines 28 and onwards, you argue a lack of literature looking at dynamic (GMM) analysis of the role of female on CSR and ESG issues, but fail to mention some examples. Please provide some, and position your work in that (brief) strand of literature. One example you should consider is Gaio, C., & Gonçalves, T. C. (2022). Gender diversity on the board and firms’ corporate social responsibility. International Journal of Financial Studies10(1), 15.

2- Please revise if it makes sense to present your research hypothesis in the introduction.

3- Please reconsider the use of adjectives such as "complicated" (line 79), interestingly (line 88) and so on.

4- What do you mean by "in the crowd" (line 141). Please revise carefully english for clarity.

5- Please revise your data: your max tangibility is over 2 (which means that Net PP&E, a subset of Assets, is 2 times those same Assets - this should be a typo. Similar problems might exist for a negative tobin's Q.

6 - In line 192 you mention "all the sectors". But you are working one sector only, aren't you?

7- Explain your sentences in lines 242-244. What did you do specifically?

8- Lines 330 and onwards are presenting conclusions that are not warranted by the results and the english is a bit strange (what do you measnby "households' anxiety"?). Similar problems stem from the following paragraphs discussing the results.

Thank you for your paper.

Kind regards
